

# *RehaBEElitation*: the architecture and organization of a serious game to evaluate motor signs in Parkinson's disease

Luanne Cardoso Mendes[1,2], Angela Abreu Rosa de Sá[3], Isabela Alves Marques[1,2], Yann Morère[2] and Adriano de Oliveira Andrade[1]

[1] Centre for Innovation and Technology Assessment in Health (NIATS), Faculty of Electrical Engineering, Federal University of Uberlândia, Universidade Federal de Uberlândia, Uberlândia, Minas Gerais, Brazil

[2] Laboratoire de Conception, d'Optimisation et de Modélisation des Systèmes (LCOMS), Université de Lorraine, Metz, Moselle, France

[3] Assistive Technology Laboratory, Faculty of Electrical Engineering (NTA), Universidade Federal de Uberlândia, Uberlândia, Minas Gerais, Brazil

Corresponding author
Luanne Cardoso Mendes, luannecmendes@gmail.com

## ABSTRACT

**Background**. The use of serious games (SG) has received increasing attention in health care, and can be applied for both rehabilitation and evaluation of motor signs of several diseases, such as Parkinson's disease (PD). However, the use of these instruments in clinical practice is poorly observed, since there is a scarcity of games that, during their development process, simultaneously address issues of usability and architectural design, contributing to the non-satisfaction of the actual needs of professionals and patients. Thus, this study aimed to present the architecture and usability evaluation at the design stage of a serious game, so-called *RehaBEElitation*, and assess the accessibility of the game.

**Methods**. The game was created by a multidisciplinary team with experience in game development and PD, taking into consideration design guidelines for the development of SG. The user must control the movements of a bee in a 3D environment. The game tasks were designed to mimic the following movements found in the gold-standard method tool—Movement Disorder Society-Unified Parkinson's Disease Rating Scale (MDS-UPDRS)—for the assessment of individuals with PD: hand opening and closing, hand extension and flexion, hand adduction and abduction, finger tapping, and forearm supination and pronation. The user interacts with the game using a wearable interface device that embeds inertial and tactile sensors. The architecture of *RehaBEElitation* was detailed using the business process model (BPM) notation and the usability of the architecture was evaluated using the Nielsen-Shneiderman heuristics. Game accessibility was evaluated by comparing the overall scores of each phase between 15 healthy participants and 15 PD patients. The PD group interacted with the game in both the ON and OFF states.

**Results**. The system was modularized in order to implement parallel, simultaneous and independent programming at different levels, requiring less computational effort and enabling fluidity between the game and the control of the interface elements in real time. The developed architecture allows the inclusion of new elements for patient status monitoring, extending the functionality of the tool without changing its fundamental characteristics. The heuristic evaluation contemplated all the 14 heuristics proposed

by Shneiderman, which enabled the implementation of improvements in the game. The evaluation of accessibility revealed no statistically significant differences ($p < 0.05$) between groups, except for the healthy group and the PD group in the OFF state of medication during Phase 3 of the game.

**Conclusions**. The proposed architecture was presented in order to facilitate the reproduction of the system and extend its application to other scenarios. In the same way, the heuristic evaluation performed can serve as a contribution to the advancement of the SG design for PD. The accessibility evaluation revealed that the game is accessible to individuals with PD.

## INTRODUCTION

The use of serious games (SG) has received considerable attention in the health domain (*Adcock et al., 2020*). SG are digital games developed to entertain players and also to achieve an additional goal, such as acquiring learning or improving the users' health (*Dörner et al., 2016*; *Ritterfeld, Cody & Vorderer, 2009*). Because exercises through games are perceived by the individuals as entertainment rather than therapy, these instruments promote more motivation to patients and reinforce adherence to the therapeutic process, contributing to the maintenance and/or improvement of the quality of life of the individuals (*Tarousi et al., 2021*). In addition to allowing movement training for motor rehabilitation purposes, which is the most commonly found application in the literature for SG (*de Oliveira et al., 2021*), patient monitoring and motor symptom assessment of a wide range of diseases, such as Parkinson's disease (PD), can be performed using SG (*Wilkinson et al., 2018*).

PD is a neurodegenerative disorder that affects approximately 2% of the population over 60 years old (*Singh & Ganley, 2021*) and is characterized by the loss of dopaminergic neurons, which promote the appearance of motor symptoms, such as bradykinesia and tremor (*Balestrino & Schapira, 2020*). Due to the variety and severity of the symptoms of this disease, the importance of implementing technological resources such as SG as a valuable tool for clinical assessment and patient-centered monitoring is highlighted (*de Oliveira et al., 2021*; *Tarousi et al., 2021*). When used for clinical assessments, SG allows health professionals to provide more personalized follow-up to patients while also assisting them in disease monitoring (*Tarousi et al., 2021*).

The objective assessment of motor signs of PD using wearable sensors has been widely studied. *de Oliveira Andrade et al. (2020)* observed that it is possible to study resting hand tremor and postural tremors using a single inertial measurement unit (IMU) placed on the back of the hand. *Rabelo et al. (2017)* discriminated groups of PD patients from healthy elderly by an objective assessment of bradykinesia using inertial sensors. However, objectively assessing such symptoms using SG may be an efficient and promising alternative, due to the advantages provided by these instruments (*Dias et al., 2016*; *Tarousi et al., 2021*).

One of the reasons that explain the scarcity of tools effectively used in clinical practice, not limited to controlled environments, is the lack of projects that actually meet the needs of professionals and patients. It is precisely in order to fill this gap that this study presents the design and implementation of a serious game, so-called *RehaBEElitation*. The game was designed considering a long experience of our research group in the evaluation of motor signs of PD (*de Oliveira Andrade et al., 2020*; *de Oliveira et al., 2021*; *Escudeiro et al., 2021*; *Folador et al., 2021b*; *Folador et al., 2021a*; *Luiz et al., 2021*; *Machado et al., 2016*; *Peres et al., 2021*; *Rabelo et al., 2017*).

A literature review conducted by our group in 2021 showed that most studies related to the development of SG for PD evaluation (*de Oliveira et al., 2021*) do not concurrently address usability issues and game architecture design. Thus, this research presents in a detailed way the game architecture design through business process model notation (BPMN), which uses a set of graphical representations to facilitate the understanding and implementation of the game architecture (*Carvalho et al., 2015*). In addition, once the game architecture is defined, it is relevant to validate it still at a design stage. This is an important step that we found to be neglected in most studies (*Andrade Ferreira et al., 2020*; *Avola et al., 2018*; *Cai et al., 2021*; *Oña et al., 2018*; *Sánchez-Herrera-Baeza et al., 2020*; *van de Weijer et al., 2019*).

Thus, we proposed using the heuristic evaluation based on the Nielsen-Shneiderman heuristics to evaluate the system's usability (*Zhang et al., 2003*). Heuristic evaluation is typically used after the system has been fully developed. In this study, we did an evaluation while the design is still in progress to predict and avoid usability issues. This is especially important in the development of SG for severe diseases such as PD, because a lack of understanding of potential usability issues may affect the patient's experience, contributing to clinical follow-up failure.

Proving that a serious game is accessible is also crucial for ensuring the quality of the game design and for enhancing the player experience. Accessibility generally refers to the elimination of barriers that prevent individuals with various disabilities from accessing or utilising a product, such as a serious game (*Fortes et al., 2017*). According to *Garber (2013)*, approximately 2% of the population, or approximately 6.3 million individuals, are unable to play computer games due to a disability. Consequently, this attribute was also considered in this study.

In relation to other studies (*de Oliveira et al., 2021*) the developed serious game is innovative in the following aspects: (i) detailed provision of the game architecture through business process model diagrams; (ii) usability evaluation at the design stage through heuristic evaluation; (iii) alignment between clinical and game design requirements; (iv) design of many events that enable the evaluation of clinical signs of PD, such as tremor and bradykinesia, at specific moments of the game; (v) development of a human machine interface (HMI) device (glove) specific for this application; (vi) proposal of a game narrative that meets the efforts required to deal with the challenges of PD; (vii) possibility of longitudinal evaluation of the disease, which ensures long-term follow-up of patients; and (viii) incorporation of the most relevant movements for clinical evaluation of hand motor signs in PD into the game.

## BACKGROUND

To determine the current state of research in this field, a literature review was conducted, which included recent studies on the use of SG for rehabilitation or motor sign monitoring of individuals with PD. A table showing the current studies has been elaborated (Attachment S1), the following characteristics are presented: the objective of the game, the required player movements, the HMI used, the symptom/characteristic evaluated, the instrument used to evaluate the game's usability and information on the game architecture design.

No study focused on the monitoring of any motor sign of Parkinson's disease through the use of the serious game. In addition, the studies did not describe the incorporation of game components that would allow objective assessment of the disease's motor symptoms. Thus, the PD symptoms were not evaluated using the serious game, but rather other instruments and/or subjective scales (*e.g.*, Box and Blocks Test (BBT), Purdue Pegboard Test (PPT), Berg Balance Scale (BBS), Parkinson's Disease Questionnaire (PDQ-39), MDS-UPDRS and Hoehn and Yahr (HY) scale). Our proposal was to develop a serious game that can be used for both rehabilitation and monitoring symptoms of PD through game interaction and recording of inertial data.

No study developed a proprietary HMI for serious game user interaction. The development of a specific instrument, as done in this study, can be beneficial in several ways, including: efficient detection of the game's movements, ease in solving calibration problems, greater customization capacity (*e.g.*, size, colours, *etc.*) to meet various demands, marketability, and a reduced cost HMI device. Several instruments were used in the studies, such as Leap Motion Controller (*Avola et al., 2018*; *Cemim et al., 2022*; *Fernández-González et al., 2019*; *Foletto, d'Ornellas & Prado, 2017*; *Oña et al., 2020*; *Oña et al., 2018*; *Sánchez-Herrera-Baeza et al., 2020*; *Shah et al., 2019*), Microsoft Kinect (*Avola et al., 2018*; *Cikajlo et al., 2018*; *Nuic et al., 2018*; *Silva et al., 2017*), HTC Vive (*Chen et al., 2020*; *Stanica et al., 2020*), Oculus Rift (*Oña et al., 2020*; *Sánchez-Herrera-Baeza et al., 2020*), Beyond Your Motion (*Blanc et al., 2022*; *Leblong et al., 2017*), Myo Gesture Control Armband (*Stanica et al., 2020*), Inter Real Sense camera (*Bevilacqua et al., 2021*), head mounted display (*Avola et al., 2018*), tablet (*Dauvergne et al., 2018*) and stepping mat (*Yuan et al., 2020*).

Some studies did not detail the usability assessment instrument, while others used questionnaires developed by the research team. No mention is made of whether such instruments were tested and validated, and no study evaluated game usability during the design stage.

Most studies did not present information related to the game architecture. Some studies mentioned the main modules and components of the game, but did not present detailed and schematized information about the architecture and organization of the system. The lack of illustrative tools, such as diagrams and schemes, may hinder the understanding about the software framework as a whole. In this study, the system was developed in a modularized way in order to implement parallel and simultaneous programming at different levels. The system modules are completely independent from each other, allowing the fluidity of the game, the control of the interface elements in real time, the inclusion of other elements and the expansion of the game functionality without changing its basic

characteristics. These game architecture features avoid its obsolescence and enable future advances.

An important consideration is that, with the exception of *Shah et al. (2019)*, no reviewed studies have assessed the accessibility of the game. This is a relevant assessment as it checks whether the disabled player is able to play the game similarly to a healthy individual, *i.e.,* it assesses whether the game can be played at a competent level by anyone.

In summary, the main contributions and novelties of this study are: (i) development of a serious game capable of assessing PD in an objective way, *i.e.,* the game itself presents features that allow the assessment of motor symptoms of the disease. Thus, the game is not only useful for rehabilitation, but also for the evaluation of motor symptoms of the disease; (ii) evaluation of the game usability at a design stage, which can contribute to a significant reduction of errors and failures in the system. This can accelerate the game refinement stage after a pilot test; (iii) careful detailing of the game architecture in the form of easy-to-understand diagrams, in order to facilitate the comprehension of the developed software framework; (iv) implementation of the system in a modularized way in order to allow future inclusion of processes and features and/or changes in the software; and (v) evaluation of the accessibility of the game to ensure that individuals with PD are able to interact with the game.

## MATERIALS & METHODS

### Design principles of a serious game for rehabilitation and symptom assessment

The guidelines and design principles presented by (*Paraskevopoulos et al., 2014*; *Shi & Shih, 2015*) for the development of serious games used for rehabilitation and assessment of motor symptoms in people with PD were taken into account:

- Inclusion of a person with PD in the game refinement process to create a user-centered tool that actually meets the needs of patients while saving time and resources;
- Incorporation of specific motor assessment exercises for PD patients into the game;
- Presentation of a narrative that uses appealing graphics and scenarios, charming animations, and amusing sounds to provide playful experiences and increase player engagement;
- Presentation of clear and objective game instructions so that players understand exactly what they need to do to achieve the objectives of the phases;
- Inclusion of visual and auditory stimuli capable of providing the player with a playful and pleasurable experience while also aiding cognitive training;
- Automatic adaptation and calibration for each player to adjust the range of movement and exercise level required by the game, providing the best experience possible;
- Provision of player progress at each stage and in each game session to encourage competition with oneself and, as a result, motivation; and
- Use of a commercially available and low-cost interface device for capturing player moves.

## The *RehaBEElitation* serious game

*RehaBEElitation* is a serious game based on bees, and it was developed for both rehabilitation and monitoring of individuals with PD. The bees represent hard work and dedication, characteristics highly demanded of PD patients during their rehabilitation process. A multidisciplinary team from the fields of biomedical engineering, computer science (with experience in game design), physiotherapy and physical education collaborated in the design and development of the game. In addition, suggestions from a PD patient were considered.

The user must control the movements of a bee in a 3D environment. The game tasks were designed to mimic the following movements found in the gold-standard method tool—Movement Disorder Society—Unified Parkinson's Disease Rating Scale (MDS-UPDRS Part III) (*Goetz et al., 2008*)—for the assessment of individuals with PD: hand opening and closing, hand extension and flexion, hand adduction and abduction, finger tapping, and forearm supination and pronation.

The game presents four phases (Attachment S2), each representing a bee worker's task in the real world. So that players understand the goal of each phase, four short videos were developed to explain to players the four tasks (each video for each task). Each video is executed before the beginning of each phase, with the option to interrupt it at any time for immediate start of the game. The objectives of the phases are:

*Phase 1:* Pollinating the flowers—The objective is to collect pollen from one flower and pollinate another one. Flowers that have pollen to collect are indicated by yellow arcs around them, and those that need to be pollinated are indicated by green arcs. The player must move the bee to a flower containing pollen and close the hand to catch it. Then, with the hand closed, the player must move the bee to a flower that does not have pollen and open the hand to deposit it.

*Phase 2:* Feeding the larvae—The aim is to feed the larvae. To move the bee up and down the player must perform the movements of extension and flexion of the hand; and to move the bee to the left and right the player must perform the movements of adduction and abduction, if playing with the right hand. The bee moves through the scenario only if the player's hand is closed. When the player places the bee in front of a larva and opens his hand, the larva is fed.

*Phase 3:* Collecting the nectar—The objective is to collect nectar from flowers. Flowers that have nectar are indicated by drops of water. The player must guide the bee to a flower that has nectar and perform the finger tapping movement to collect the nectar.

*Phase 4:* Drying the nectar—The aim is to dry the nectar in order to produce honey. The player must move the bee to the alveoli of the hive containing nectar and perform the forearm supination and pronation movements. The alveoli containing nectar are indicated by light reflections. The required movements enable the bee to flap its wings more quickly and dry the nectar.

There are nine different difficulty levels in the game. The characteristics that change according to the level of difficulty are the number of targets that should be reached by the player, the maximum time of the phase, and the speed of the bee movement. The duration

times of the phases in each difficulty level were determined to respect minimum conditions for playing the game, and, at the same time, to challenge the players.

Each correct hit adds 10 points to the score. It is important to highlight that the player can evolve in the difficulty level independently, *i.e.,* can increase the difficulty level in certain phases and not in others. The difficulty level is advanced for phases where the player reaches the objectives in the established time and maintained for those that this does not occur.

Brainstorming sessions were held to ensure that the game idea was accurately translated into the game design. These sessions were repeated, and the game design refined until a prototype was ready to be evaluated. The usability of the SG with PD patients was evaluated, and the results showed that *RehaBEElitation* is a game that presents simple and intuitive narrative and interface, compatible with the players' mental models. This facilitated the interaction of individuals with the game and contributed with the great acceptance of the whole system (*Mendes et al., 2022b*).

## Interface device

To enable the communication between the game and the real environment, a human machine interface (HMI) was developed. It consists of a glove composed of inertial sensors, located in a box attached to the glove, on the back of the hand; and capacitive sensors, conductive thread sewn into the glove, towards the palm of the hand and the fingers (*Rosa et al., 2021*).

The HMI is composed by a microprocessor (ESP32) which communicates with an inertial sensor (MPU6050) *via* the I2C protocol. The inertial sensor is composed of an accelerometer and a gyroscope (MPU6050), and a magnetometer (QMC5883L). The MPU6050 has a 3-axis accelerometer and a 3-axis gyroscope. The fusion of accelerometer and gyroscope data is processed by the digital motion processor (DMP) to obtain quadrature orientation. The magnetometer (QMC5883L) has been integrated into the system to implement an electronic compass that merges the values of the accelerometer and the magnetometer to produce north, *i.e.,* a reference angle based on the geomagnetic field. Therefore, this north is used to correct the error in the estimated angle around the $z$ axis.

The DMP configuration (on MPU6050) has been defined with a sensitivity of $\pm2g$ (accelerometer) and $\pm2,000dps$ (gyroscope), a low-pass digital filter of 184 Hz (accelerometer) and 188 Hz (gyroscope). The digital low pass filter is intrinsic to the DMP and the higher bandwidth was chosen for this first approach. For external magnetometer, the sensitivity was set to $\pm8G$. The sampling rate was set to 100 Hz (*Rosa, 2022*).

Quaternions are used to obtain pitch and roll angles, and this information is aggregated *via* the user datagram protocol (UDP) for communication and transmission from the HMI to a computer running the serious game over a Wi-Fi link. The operation of the glove was shown in a schematic diagram (Attachment S3).

The device was validated through accuracy, precision and performance tests, and was shown to be a feasible tool for collecting motor signals from PD patients (*Rosa, 2022*).

There are 12 different states based on the data supplied by the glove to the game, each of which is related to the bee movement in the game. The relationships between states, hand movements and bee movements in the game are shown in (Attachment S4).

## Clinical assessment using the _RehaBEElitation_ serious game

The serious game can be used to evaluate motor symptoms of PD, such as bradykinesia and tremor. Assessing these symptoms during a game session is an excellent alternative to monitor health status and disease progression of patients. For this, in addition to using signals collected by the HMI employing inertial measurement units, some markers were defined and incorporated into the game. Such markers are represented by events, responsible for characterizing specific occurrences. Totally, 23 events were defined, shown in supplementary material (Attachment S5).

The bradykinesia, for example, can be evaluated by calculating the duration of the event 17 (_Mendes et al., 2022a_). This event starts at the instant when the player reaches the target, _i.e.,_ when visual and sound stimuli are presented to the player indicating that he can start the movement execution required to score points. The event ends at the instant when the player finishes the movement execution and consequently scores points in the game.

Some movements, such as opening and closing the hand, can help detect motor fluctuations caused by PD. The detection of these fluctuations is important clinically because a high incidence of motor fluctuations may indicate poor treatment efficacy or non-motor problems (_e.g._, depression, stress, and anxiety) that are common in PD patients.

## Implementation of the architecture

Unity 3D was used to create the scenarios and interactions, and Blender 3D for the modelling of the game objects. The control panel was developed using Visual Studio 2019; and the database, using PostgreSQL. An overview of the general modules of the serious game is presented in (Attachment S6). The glove detects hand movement, and a commercial wristband collects physiological signals (E4—Empatica). The data are synchronized to ensure the correct evaluation of the disorder's motor and non-motor symptoms.

## Usability evaluation

Heuristic evaluation is an evaluation technique used to identify major usability problems of a system that can be employed at a design stage. This technique requires three or more evaluators to apply usability heuristics to the system to identify violations and assess the severity of each violation (_Zhang et al., 2003_). The usability was evaluated by four evaluators with experience in game development, computer science, biomedical engineering and PD. The severity scores were defined as: 0—No usability problem; 1—superficial problem. Does not need to be fixed, unless extra time is available; 2—minor usability problem (low priority to fix); 3—major usability problem (high priority); and 4—usability catastrophe. Must be fixed prior to implementation.

As a result, a table was created detailing how the heuristic was contemplated in the project, besides presenting the description of the usability problem that could be generated if the heuristic was not satisfied, and the severity of each violation.

## Game accessibility evaluation

An experimental session was conducted to assess the game accessibility. This was performed to assess whether the impaired group can play as well as the healthy group, *i.e.,* whether the serious game provides accessible gameplay. In total, 30 participants were involved, being 15 healthy subjects and 15 individuals with PD, matched in age and gender. Data were collected as previously described in (*Mendes et al., 2022a*). All participants played the game in the easiest difficulty level. The PD group interacted with the game in both ON and OFF medication states. The ON state occurs when disease symptoms are under medication control, and the OFF state occurs when symptoms are not being properly controlled by medication (*Capriotti & Terzakis, 2016*; *Lees, 1989*). The study was approved by the Ethics Committee of the Federal University of Uberlandia under the CAAE number 39187620.6.0000.5152.

The game accessibility was assessed by estimating the final score of each game phase. In this way, it is possible to compare the performance of players in each group, and verify whether players with PD have similar ability to play the game as healthy players.

One-way analysis of variance (ANOVA) was used to compare group performance. When the ANOVA assumptions were not met, the non-parametric Kruskal–Wallis test was used, with Wilcox test with Bonferroni correction used for pairwise comparisons between groups. A $p$-value of 0.05 was used in all cases. Data analysis was performed in R (*R Core Team, 2020*).

## RESULTS

### The architecture of the *RehaBEElitation* serious game

The general BPM describing the interaction between the actors of the system (patient/player and therapist/researcher) and the game processes is shown in (Attachment S7). Figure 1 shows the BPM for the calibration of the game. Figures 2 to 5 depicts the BPM from Phase 1 to 4 of the game. Figure 6 shows the BPM related to the data collection stage.

### Heuristic evaluation of the *RehaBEElitation* serious game

Table 1 presents the usability evaluation of the game at a design stage. The severity scores are based on the mean score estimated from four evaluators (designers) specialized in game designing, PD and usability assessment.

### Game accessibility evaluation

The main characteristics of the participants are given in (Attachment S8). ANOVA was used to determine whether there was a difference in age between the healthy and PD groups. The test result confirmed that there was no statistical difference in age between the groups ($p > 0.05$). The Kruskall–Wallis rank sum test was used to compare MDS-UPDRS III scores in the ON and OFF states of medication, and it revealed a statistical difference ($p < 0.05$) between the experimental conditions (ON and OFF). ANOVA was used to compare Hoehn and Yahr scores in both states of medication and found no statistical difference ($p > 0.05$). Figure 7 shows the scores of the individuals for each group in each of the four phases of the game.
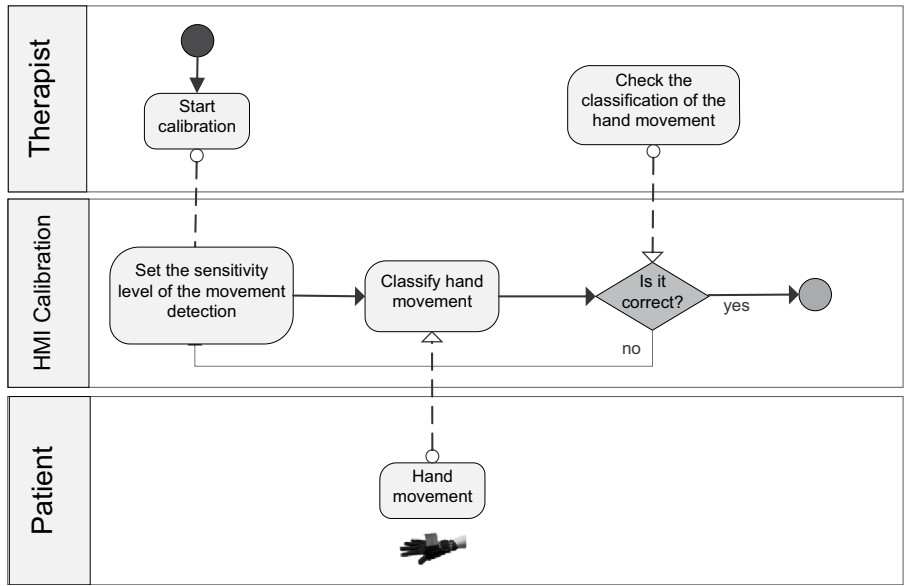

**Figure 1** BPM representing the calibration process of the game.

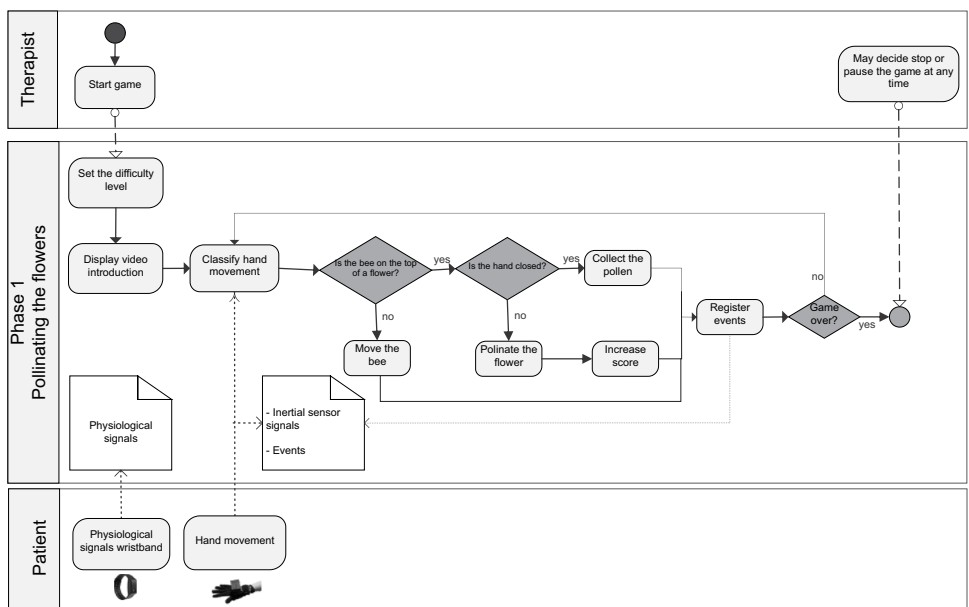

**Figure 2** BPM representing the process of the Phase 1 of the game.

In Phases 1, 2, and 4, no statistically significant differences ($p < 0.05$) were found, indicating that the scores of the participants were similar, suggesting that in these phases, all players are able to meet the phase objectives equally. In Phase 3, statistically significant differences in scores were found between groups of healthy participants and those with PD in the OFF state ($p = 0.2761$). These findings suggest that the movement required in

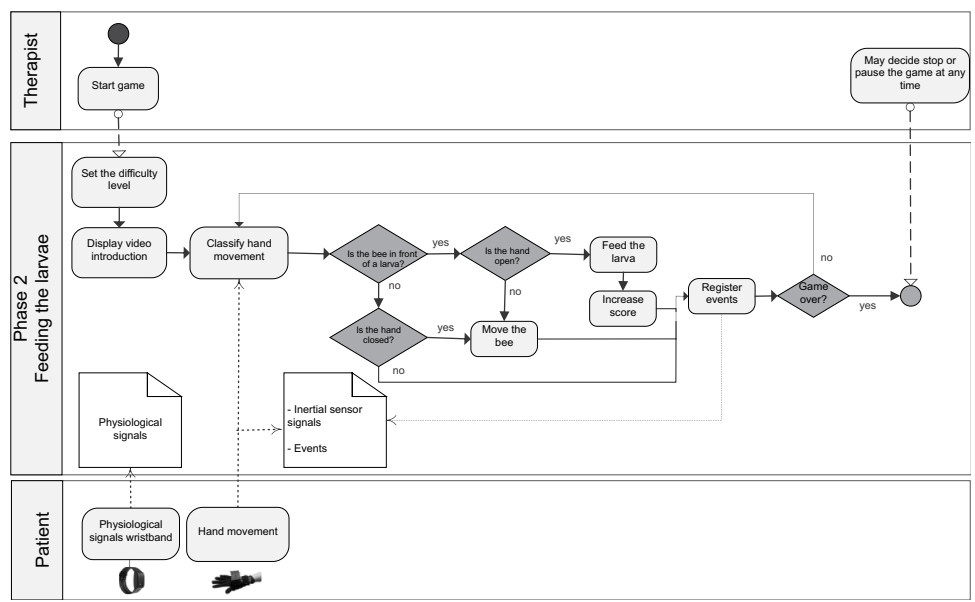

**Figure 3** BPM representing the process of the Phase 2 of the game.

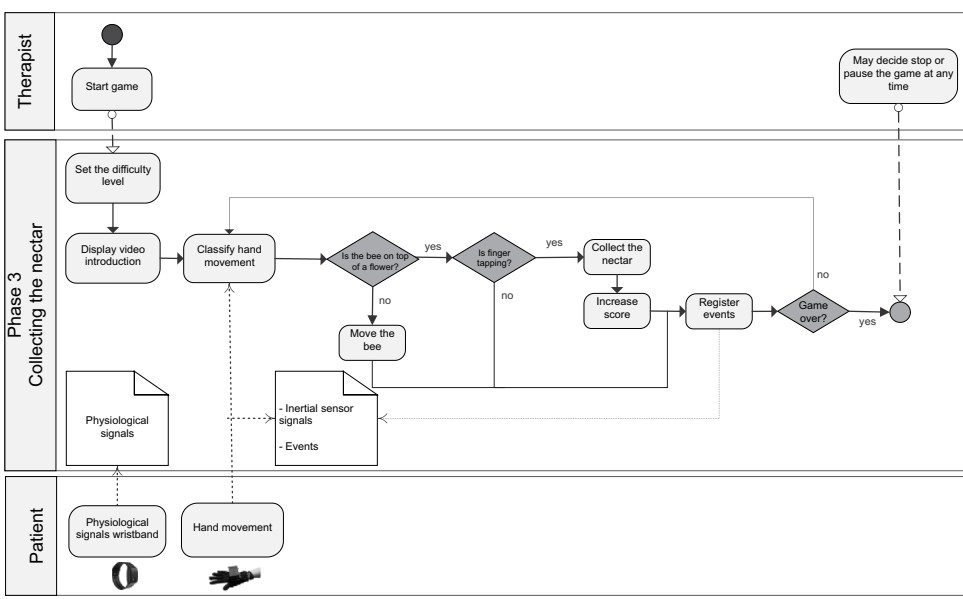

**Figure 4** BPM representing the process of the Phase 3 of the game.

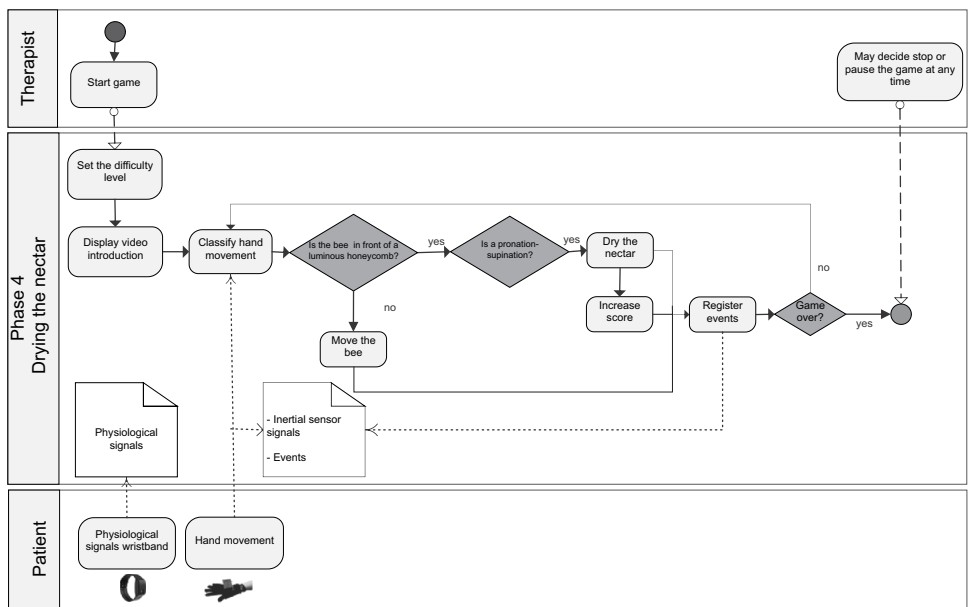

**Figure 5** BPM representing the process of the Phase 4 of the game.

Phase 3 (finger tapping) is difficult for PD patients in the OFF state to perform because it requires fine dexterity of the hand's fingers.

# DISCUSSION

The serious game *RehaBEElitation* was created to meet a real demand in the evaluation of upper limb motor signs in PD. The hand plays a fundamental role in our functional activities and its function can be severely affected in early stages of the disease, being of great relevance to evaluate motor signs in this region. Although there are initiatives in the literature for the assessment of upper limbs in PD using SG, there is a lack of development of methods that insert protocols from consolidated clinical tools, such as the MDS-UPDRS, in the game design. This is undoubtedly one of the great differentials of this study. By incorporating knowledge of clinical practice to the game, the result of the game will be more easily accepted by the clinical community and the patient will have an easier interaction with the game since he is used to perform certain movements during clinical evaluation sessions.

By incorporating a wearable technology to monitor and record motor signs of PD, the healthcare professional or researcher has at his disposal a tool to visualize events that are imperceptible to the unaided eye, complementing and expanding the usual clinical evaluation. Thus, using sensors that allow the professional to identify nuances about the disease may be important, and may even be the differential factor for identifying the beginning of a serious disease or a change of trajectory in the disease treatment. From this perspective, *i.e.,* a real demand to accurately monitor some PD symptoms (bradykinesia, tremor or motor fluctuations), the *RehaBEElitation* emerges as an instrument that can be

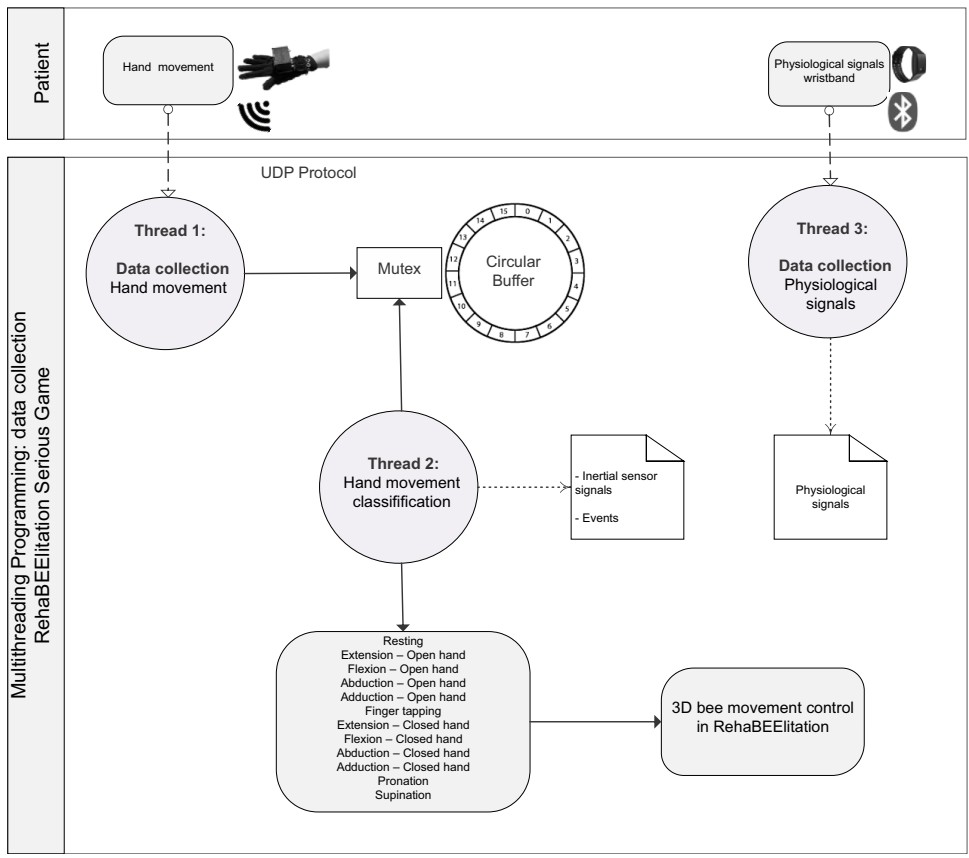

**Figure 6** **BPM representing the process of data collection.**

exactly useful in the clinical and even surgical environment, *e.g.*, it can be used to evaluate the results of ablative surgeries and deep brain stimulation.

Several studies (*Balci et al., 2021*; *Falla et al., 2021*; *Lubomski, Davis & Sue, 2020*; *Montanaro et al., 2022*) point to a high incidence of depression, anxiety and other negative emotional aspects that accompany PD. This contributes to the patient not having the necessary engagement in rehabilitation processes or even clinical assessment. Thus, the use of a playful assessment scenario, such as the one provided by the game, can be the difference between a successful or an inaccurate assessment. It is known that the mere fact that a person with PD is being monitored can instantly affect their motor signs. Thus, if the person is immersed in the virtual reality provided by the game, the influence of external factors can be reduced, contributing to a more accurate assessment of the effectiveness of treatments and the clinical condition of the individual.

The requirements gathering stage for the development of the *RehaBEElitation* serious game considered the participation of patients and professionals with experience in PD (*e.g.*, biomedical engineers, physiotherapists, physical educator, researchers, neurologist, neurosurgeon, and experts in computer science, ergonomics and human factors). Unlike several existing studies in the literature (*Cikajlo et al., 2018*; *Fernández-González et al., 2019*;

**Table 1** Usability evaluation of the game at a design stage.

| Heuristics | How the heuristic was contemplated in the project | Description of the usability problem | Severity |
|---|---|---|---|
| **H1. Consistency and patterns** - Users should not have to worry about whether words, situations, or actions mean the same thing. | 1—The interface for game control was designed considering ergonomics and the positioning of sensors in pre-established regions for signal capture during clinical exams. The consistency consists of following the positioning already adopted in clinical studies. | 1—The health professional (researcher) has a mental model that comes from training. Changing the positioning of the sensor can generate confusion about the information acquired. | 2.50 |
| | 2—Perform game calibration considering clinical routine, clinical experience and patient conditions. | 2—Do not ensure that the movement of the bee follows the movement of the hand, generating stress, demotivation, frustration, non-understanding of the activity. | 3.50 |
| | 3—Alignment between the tasks of the avatar (bee) with the tasks that the bees perform in the real world. For example, the bee pollinates in the real world as well as in the game. | 3—Problems interpreting the narrative of the game because it violates a mental model established according to the experiences of the participants about the role of bees in nature. | 2.00 |
| | 4—Direct relationship between the evaluations required in the clinic with those required for game control. | 4—Patients would not be stimulated to perform movements required for clinical assessment. | 4.00 |
| **H2. Visibility of system state** - The system should keep the user informed of what is occurring, through appropriate feedback. | 1—After the system calibration stage, with the objective of correctly identifying the hand movements, the system presents to the user the state in which the player's hand is, offering feedback to the user that the interface is functioning correctly. | 1—Difficulty in identifying problems related to game control. For example, the user (player) could move his hand to the right and the avatar does not obey the instruction given by the user. | 3.25 |
| | 2—Availability of real-time information showing that the physiological and inertial signals were effectively being collected. | 2—It may be impossible to control the avatar. | 2.00 |
| | 3—A light signal was available on the glove indicating that the device was on. | 3—It may be unable to control the avatar. | 2.00 |

**Table 1** (*continued*)

| Heuristics | How the heuristic was contemplated in the project | Description of the usability problem | Severity |
|---|---|---|---|
| | 4—The targets were identified by the presence of markings such as arcs and light reflections to facilitate target identification and correct avatar positioning. | 4—As the game was built in three dimensions, the target location could be harmed by problems related to depth perception in the virtual scenario. | 3.75 |
| | 5—Presentation of a virtual object (hand in 3D) that showed the player the movement needed to perform the expected task. | 5—The player might not understand which movement should be done to execute the task. | 3.00 |
| | 6—Presentation of the players' score progress by a progress bar that is filled with honey. | 6—Misinformation about the game progress. | 1.25 |
| | 7—Visual and auditory feedback whenever the player reached the target, performed the correct movement, and met the phase's objectives. | 7—Loss of engagement of the player during the game. | 2.25 |
| **H3. Correspondence between the system and the real world** - The system must "speak" the same language as the user, making information natural and logical. | 1—Correspondence between the required task and the intuitive movement that the hand performs. In the real world the movement of closing the hand is intuitively associated with collecting things, and opening the hand is associated with depositing things. Then, in the real world the player closes his hand to collect the pollen and opens his hand to deposit it. | 1—Reduce player engagement, since using random, non-intuitive movements could interfere with the player's mental model and expectation. | 3.00 |
| | 2—The game narrative corresponds to the tasks the avatar performs in the real world, such as pollinating flowers, feeding larvae, collecting and drying nectar. | 2—Reduce player engagement, since using random scenarios or activities could interfere with the player's mental model and expectation. | 2.75 |
| | 3—The movements used in the game are in accordance with those performed during clinical assessments, such as hand flexion and extension, finger tapping, pronation and supination, opening and closing of the hand. | 3—Ineffectiveness of the game for movement rehabilitation and motor symptom assessment in PD. | 4.00 |
| | 4—Insertion of the avatar in the expected environment in the real world, e.g., the avatar (bee) was inserted in scenarios such as garden with flowers and inside a hive. | 4—Reduce player engagement, since using random scenarios or activities could interfere with the player's mental model and expectation. | 2.50 |
| | 5—Correspondence of the virtual elements of the game, i.e., pollen, nectar and larva, with their actual form found in the real world. The pollen in the game is represented by grains of the same form that is found in nature. The nectar is represented by water since in the real world 70% of its composition is water. The larva is represented by a real image of a bee larva. | 5—Difficult the player's immersion. | 2.50 |

**Table 1** (*continued*)

| Heuristics | How the heuristic was contemplated in the project | Description of the usability problem | Severity |
|---|---|---|---|
| **H4. Minimalist design** - Dialogs should contain only the necessary information. | 1—Avoid excessive use of engagement reinforcement through virtual elements. In a preliminary version of the game, two virtual elements were incorporated, a butterfly and a bird, to reinforce the player's motivation. The butterfly appeared in the scenario whenever the player remained more than 30 s without making any movements. The bird would appear whenever the player did not score for about a minute. However, after a preliminary test, the designers verified that such elements were excessive, producing negative stimuli (*i.e.*, irritation by the constant presence of stimuli). | 1—Increased stress and anxiety during the game, reducing immersion in the game. | 2.00 |
| | 2—Presence of few visual elements in the scenario. In phases 1 and 3 of the game the scenery is composed by flowers, grass, some insects and the presence of a single tree with a beehive. In phases 2 and 4 the game takes place inside the beehive that has only the honeycomb with the targets (the larvae or the nectar). | 2—The increase of the number of elements in the scenario could cause distraction to the player, making it difficult to reach the game objectives. | 1.00 |
| **H5. Recognize rather than remember** - Users should have no need to remember information from one part of the system to another. | 1—The use of visual and auditory feedback to recognize relevant steps of the game. For example, when the user reaches the end of the scenario, a virtual hand is shown indicating the movement that should be performed by the player to continue the game. | 1—The user could imagine that the game was terminated early. The user might not execute the correct move at the correct time. | 2.50 |
| | 2—Movements already known and practiced during clinical assessment were incorporated into the game, thus facilitating recognition and intuitive execution of the movements. | 2—Overload the gameplay process, because besides learning the game rules, the player would have to learn how to execute the required moves. | 3.00 |
| **H6. Informative feedback** - Immediate and informative feedback should be given to users about their actions. | 1—Presence of visual elements in the interface that help the execution of movements at the moments in which they should occur. For example, the 3D hand indicates the movement that the player must perform when he reaches the target. In addition, the hand indicates to the player the exact way to execute the movement, effectively promoting the recording of motor signs of Parkinson's disease or improving the results of the application of the game as a rehabilitation tool. Another example is the use of target delimitations through visual elements that assist the correct positioning of the avatar. | 1—Difficulty to comprehend the game and its current state. The lack of visual elements on the interface hinders the learning process on how to actually play the game. In a preliminary version of the game, in phase 1, when the user reached the end of the scenario, a 3D hand always appeared open, indicating the adduction or abduction movement that the player should perform; however, if the player was holding the pollen (with a closed hand), when seeing the virtual open hand, the player automatically opened his hand releasing the pollen at the wrong moment. | 3.00 |
| | 2—Presence of sound feedback when the player scores, reaches a target and completes the phase. | 2—Difficulty to establish a mental model that facilitates the understanding of the game state. | 2.25 |

**Table 1** (*continued*)

| Heuristics | How the heuristic was contemplated in the project | Description of the usability problem | Severity |
|---|---|---|---|
| **H7. Flexibility and efficiency of use -** The system should satisfy both beginners and experienced users. The provision of shortcuts is a way to adjust to the various types of user. | 1—Nine difficulty levels were established with the intention of facilitating gameplay by people affected by distinct severity levels of motor signs of PD. In addition, the presence of nine levels of difficulty contributes to the development of rehabilitation protocols customized to the reality of the patient; and also to evaluate pharmaceutical treatments, since it is expected that individuals with PD who are responding more adequately to treatment will find it easier to play the game at different levels of difficulty. | 1—People with distinct motor limitations, due to the severity of motor signs such as tremor and bradykinesia, might not be able to play the game. Moreover, players might not notice their progress in the game or even not be able to interact with the game. For some volunteers the interaction could be too easy and for others too complex, interfering with the flow experience. This could further impact adherence to the rehabilitation protocol. | 3.50 |
| | 2—The calibration of the system is done in a personalized way, respecting the motor limitations of people with Parkinson's disease. This way, people with a smaller range of movement can interact effectively with the game. | 2—The control of the avatar would be impaired. The number of people able to use the system would be reduced. This may produce negative feelings in the users, as they may have the feeling of being unable to perform the activities. In addition, since the calibration is done with the execution of the movements in their maximum amplitude, the absence of a personalized calibration makes the patient's movement be performed below what he really could execute, jeopardizing the result of treatments and evaluations. | 3.00 |
| | 3—The glove project was developed to allow the free execution of movements. The glove was projected in different sizes, respecting hand anatomic patterns. The glove material was merino wool, which enabled the development of a firm glove for the connection of the box with the electronic circuitry and provided comfort to the user. In addition, a conductive thread was used to make it possible to embed the electronic circuits directly into the fabric and at the same time ensure the execution of the movements. | 3—The absence of ergonomics could cause discomfort to the players and cause the discontinuity of the therapeutic process. For example, in a first version of the game it was used a glove with thicker fabric that made it difficult to close the hand. | 2.50 |

**Table 1** (*continued*)

| Heuristics | How the heuristic was contemplated in the project | Description of the usability problem | Severity |
|---|---|---|---|
| **H8. Assist users to recognize, diagnose and recover from errors** - Error messages must be clear, objective and suggest solutions. | 1—Presence of auditory and visual feedbacks that indicate an action error caused by the player. In phase 1, when the player releases the pollen at the wrong moment the representation of the pollen on the screen disappears and a negative audio feedback is presented. | 1—The user does not notice the wrong execution of the movement and does not have the opportunity to correct the action. This, can leave him confused, affecting his learning in relation to the game. | 2.25 |
| | 2—When registering a user already in the system, an error message is presented to inform that that user is already registered. | 2—Causing duplication of the same user in the system. | 2.00 |
| | 3—An alert message informing that the data collected will not be saved if the user does not play all the phases in a single trial session. | 3—Lack of data to fully evaluate the motor signs of the person with PD. This does not guarantee that the patient will perform all the movements necessary for evaluation or rehabilitation. | 3.00 |
| | 4—Before starting the calibration process an alert message is presented to the player asking him to keep his hand in a stable position to ensure that the calibration is performed correctly. | 4—The avatar control will be affected, impairing the interaction. | 3.50 |
| **H9. Error prevention** - Preventing errors is better than providing good error messages. | 1—The data is automatically saved right after the execution of the four phases of the game. | 1—Data can be lost, impairing the assessment and recovery of patients with PD, loss of time and financial resources, discontinuation of periodic follow-up for medication adjustment and distrust of the process involved in the assessment. In cases of evaluation of patients on the OFF state of medication (when the patient's body is not under the effect of the drug) this loss of data is even more damaging since this situation produces great discomfort to the patient. | 4.00 |
| | 2—The game does not allow phases to be started without prior selection of a user and calibration of the glove. | 2—Absence of association between the registered data and the user identification. | 3.75 |
| **H10. Clear closure** - Every task has a beginning and an end. | 1—Visual and audible feedbacks are shown to the player when game tasks and phases are completed. A positive feedback message is presented whenever the player fulfills the phase objective in the established time. | 1—Lack of perception about the closure of game actions. This can cause distress, tension, irritation and discontent. | 3.00 |
| | 2—At the end of the collection session, the system informs that the data was successfully saved. | 2—Doubts can be generated about the storage and availability of the data. | 1.75 |

Cardoso Mendes et al. (2023), *PeerJ Comput. Sci.*, DOI 10.7717/peerj-cs.1267

**Table 1** (*continued*)

| Heuristics | How the heuristic was contemplated in the project | Description of the usability problem | Severity |
|---|---|---|---|
| **H11. Reversible actions -** Users should be allowed to recover from errors. | 1—Possibility of executing tasks in an indeterminate way. In phase 1, if the player cannot reach a target in the scenario, the game allows the player to naturally return to the lost target. | 1—The player does not complete the planned tasks, thus interfering on the therapeutic benefits of an effective assessment. | 3.00 |
| | 2—Users can make both correct and incorrect moves without the game interfering. In phase 1, if the player does not close his hand when collecting pollen, the game avatar remains on the target until the correct move is made. If the player makes an incorrect finger tapping movement (tapping all fingers together) in phase 3, the game avatar will remain on the target until the correct movement is made. | 2—If the game does not offer the opportunity for the player to repeat the movement, the user may become frustrated and not learn the correct movements for the use of the game. In addition, the user may make random movements that will not contribute to his clinical assessment. | 2.75 |
| **H12. User language -** Language must always be presented in a way that it is understandable by the intended users. | 1—The textual and symbolic elements are simple and understandable. When the user finishes a phase satisfactorily a short text message appears on the screen ("Congratulations"). On the other hand, when he or she finishes the game phase unsatisfactorily, the message "Time Out" appears on the screen. In addition, the 3D model of the avatar (bee) and the hand were elaborated to represent reality, being easily understood by the users. | 1—Hinder the interaction between the user and the game, causing boredom, confusion regarding performance and demotivation. | 2.00 |
| | 2—Introductory videos for each phase were elaborated in order to contextualize the game narrative. | 2—If the player does not understand the tasks of a bee in the real world, he will not understand the narrative of the game. | 2.75 |
| **H13. User control -** The system should be designed in such a way that users initiate actions, not respond to them. | 1—Perform a calibration procedure that correctly identifies the movements required for controlling the game avatar. | 1—The lack of an adequate calibration may cause the user not to dominate the game, but rather to feel dominated by the game. | 4.00 |
| | 2—Inclusion of standardized visual and audible feedback to facilitate memorization and learning of task execution. | 2—The absence of visual and sound feedback can lead to unexpected results and may result in the feeling that the user is not effectively controlling the system. | 2.00 |
| **H14. Help and documentation -** The system must provide help and have a quick search form. | 1—The game includes training phases in which the user exercises and learns the movements required to control the avatar. | 1—The absence of the training phase can make learning more difficult, hindering the interaction and interfering in the rehabilitation and monitoring of PD symptoms. | 2.75 |
| | 2—The presence of virtual objects that aid in determining the proper movements to be performed. | 2—The absence of virtual objects can have an impact on movement execution, interfering with gameplay and clinical assessment. | 3.00 |
| | 3—The game has introductory videos that explain its narrative. | 3—The absence of videos about the game narrative can cause the player to become disconnected from the game, affecting gameplay. | 2.75 |

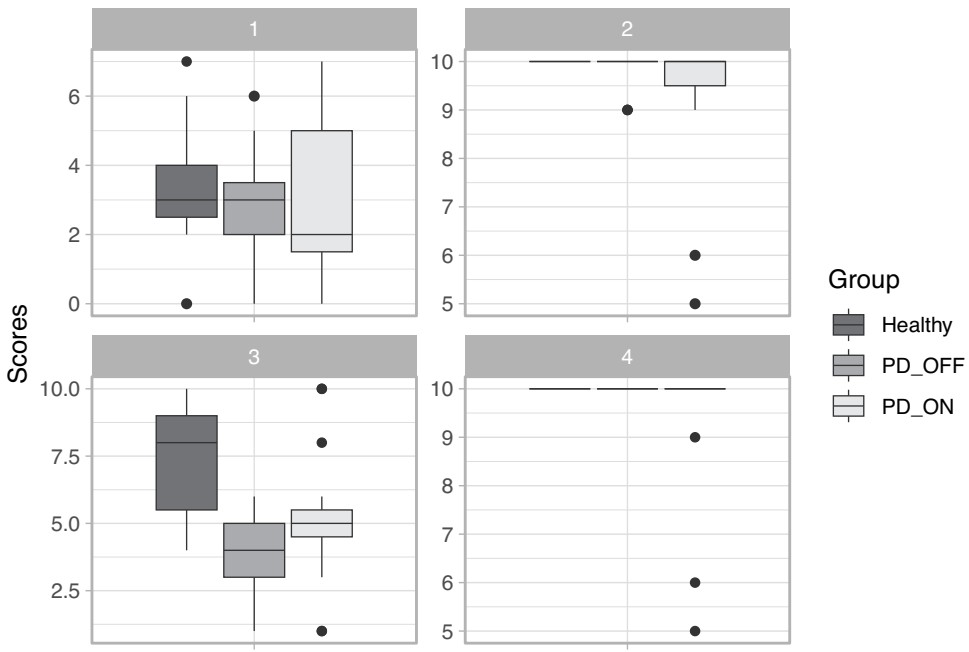

**Figure 7** Scores in each phase of the game.

*Sánchez-Herrera-Baeza et al., 2020*), the game was developed in such a way to place the end-user as the center of the development process. This may be fundamental for a good acceptability of the game. However, clinical evaluations should be conducted in the future to verify this hypothesis.

The game architecture was designed having in mind two actors: the individual with PD (*e.g.*, the player) and the professional who will be performing the assessment (*e.g.*, physical therapist, neurologist or a researcher). These actors can interact with the system at different times, and the game was modularized in order to customize interfaces for each type of actor. For instance, the therapist has exclusive access to glove calibration interfaces, user registration and experimental session customization. These modules are completely independent from other modules of the system, allowing future advances, such as the inclusion of modules for the generation of reports (*e.g.*, for the presentation of the frequency of motor fluctuations, tremor and bradykinesia) and connection with other data management systems for people with PD (*Folador et al., 2021b*).

The system architecture considers the possibility of the user carrying out training sessions before experimental and clinical sessions. The independence of these steps is essential for the user to understand the game and execute the movements correctly, since it is through the execution of such movements that the clinical assessment will be performed. At the same time, the system requires that during the clinical assessment the user performs all the processes related to the four phases of the game to ensure the execution of a protocol that allows the assessment in different contexts, as recommended by the UPDRS clinical scale.

The system was modularized in order to implement parallel and concurrent programming at different levels. For instance, the microcontroller responsible for

interfacing the glove with the game estimates all parameters independently from other running processes. This requires less computational effort, which contributes to the use of more accessible hardware, since in countries like Brazil the cost of computers is still high and modern computers are not always available in clinical environments. Furthermore, the independence between the processes contributes to the fluidity of the game and the control of the interface elements in real time.

Furthermore, the developed architecture allows the inclusion of new elements for monitoring the patient's state. For instance, in the current version of the game, there is a module which is responsible for collecting and storing data from a commercial device (*e.g.*, E4 wristband). This feature is desirable, since it extends the functionality of the developed tool without changing its fundamental characteristics. Thus, modularization also contributes to avoid obsolescence of the game, since other devices can be attached to it without compromising its basic operation. Finally, the inclusion of other sensors in the system enables the extension of the use of the game for the evaluation of new forms of motor and non-motor PD signs contemplated in its original design.

The developed system allows recording the biomedical and inertial signals of the participants. In addition, it records pre-defined events that can help the evaluation of the variability of biomedical signals during the period of the occurrence of the events. This functionality presents great clinical relevance, as it allows the assessment of the user in several contexts, for example, during the execution of different tasks and reception of different visual and sound stimuli.

The system design was developed in an incremental manner always considering the possibility of improvements due to unit test results (Fig. 6). Thus, even with the progress of the system development, new improvements were integrated into it according to the results of the usability evaluation performed at the design stage (Table 1). To evaluate the usability, the perception and experience of four evaluators were considered. In general, usability problems were identified with severity levels above 1 (that is, problems of medium, high or extremely high nature). By incorporating the necessary usability improvements to the game, we avoided that users are subjected to use a system still in an incipient stage (*e.g.*, with basic usability problems), contributing to the good acceptance of the tool in the clinical scenario.

The heuristic evaluation performed contemplated all the 14 heuristics proposed by Nielsen-Shneiderman. It was possible to identify improvements to be implemented due to each of them, emphasizing the relevance of the use of heuristic evaluation still at the design stage of SG development. In a recent literature review, which contemplated the use of SG in PD (*de Oliveira et al., 2021*), it was not found any study that presented how the heuristics that aid the development of a game with good usability were contemplated. The application of this method in this study is also a contribution to the advancement of the design of SG for PD.

From the heuristic evaluation it was possible to identify the following usability issues classified as extremely serious (severity equal to 4): (i) the absence of a calibration step, enabling the adequacy of the game control to the motor impairment of the person with PD, could substantially affect the control of the game, causing serious gameplay problems;

(ii) the meaning of a loss of data from an experimental session goes far beyond the loss of a file on a computer. This is a problem that can and should be seen as a usability problem. The loss of data may mean that the user is again exposed to a complex and long evaluation (since he may even be without the effect of medication) and that this previous experience may generate frustration and distrust in him, which totally interferes in the usability of the system. (iii) The lack of direct relationship between the evaluation performed by the game and that performed in a clinical scenario may generate problems related to the difficulty of understanding the tasks and performing movements, effectively jeopardizing the use of the game.

The game accessibility evaluation was performed to assess whether patients with PD were able to play the game in the same way as healthy patients (*Shah et al., 2019*). Statistical tests showed that the groups of healthy individuals and those with PD are homogeneous in terms of age. In general, as indicated by the MDS-UPDRS III and Hoehn and Yahr scores (Attachment S8), PD patients in the OFF medication state showed a higher degree of impairment compared to the ON state. However, despite all limitations, the assessment of the game accessibility revealed that *RehaBEElitation* can be played by individuals with PD in both medication states as well as by healthy individuals. Phase 3 was the only one that showed statistically significant differences between the scores of the healthy group with the PD group in the OFF state. As finger tapping is a movement that requires fine motor dexterity of the fingers, it is expected that a patient without the effect of a medication controlling the motor symptoms of the disease would indeed feel more difficulty in performing the movement than an individual without impairment.

Due to situations such as the one imposed by the COVID-19 pandemic, the need to develop technologies for remote use was observed. In this context, the *RehaBEElitation* serious game could be used for remote assessments; however, the way the game was implemented does not allow its use in this scenario at this time. In addition, the compliance with the heuristics for usability assessment also suggests that the game can be incorporated into clinical practice.

In the future, we intend to incorporate into the game components that also allow the assessment of lower limbs of individuals with PD, enabling, for example, gait assessment. In this way, it would be possible to perform a complete and objective evaluation of patients by using the developed system.

The system described in this article was registered with the Brazilian government (https://www.gov.br/inpi/en; INPI—registration number: BR 51 2021 001975 0) and some videos illustrating its operation are available as complementary material.

## CONCLUSIONS

This study presented the architecture and the usability evaluation at the design stage of the *RehaBEElitation* game. The user-centered system was conceived from requirements of serious game projects, having in its conception the participation of a multidisciplinary team with experience in game development and PD. The proposed architecture was presented using the BPM in order to facilitate the reproduction of the system, and to extend its application to other scenarios of use considering the evaluation of other diseases. In the

same way, the heuristic evaluation presented can serve as a guide for the development of new SG.

Some limitations of the game can be described, for instance, it cannot be applied remotely at home. This is due to the following reasons: (i) the game must be played in the presence of a professional, who can clarify doubts and observe the execution of the tasks; (ii) in several countries, such as Brazil, there is still no legal regulation that allows the use of systems of this nature at home. Thus, we chose not to incorporate this possibility of use, although the game architecture is flexible enough to allow future incorporation; (iii) the use of SG involves the registration of biomedical data, and the use of the system at home may compromise the data security; (iv) the use of games at home requires the availability of connection to stable network systems, often not available to users. Moreover, in the current version, the game cannot be played by hearing impaired patients, as they do not perceive important sound feedbacks emitted by the game. However, we are working to incorporate in the HMI device vibratory sensors to allow the delivery of tactile feedbacks to these individuals. Finally, the game cannot be played simultaneously by multiple players in the current version; however, the game architecture also enables the implementation of this feature.

In summary, the developed system allows the evaluation and monitoring of individuals with PD using an accessible serious game, objectively. The detailed description of the game architecture presented in this study, as well as the evaluation of the game usability at the design level can help professionals from different areas to develop more efficient systems and technologies.

## ACKNOWLEDGEMENTS

The authors would like to thank all the researchers and students involved in this research project (Rodrigo Ramos Rosa, Camille Marques Alves, Eduardo Lázaro Martins Naves, Fábio Henrique Monteiro de Oliveira, Marcus Fraga Vieira, Guy Bourhis, Kennedy Rodrigues Lima, Edgard Afonso Lamounier Júnior, Pierre Pino, and Adriano Alves Pereira).

### Funding

The present work was carried out with the support of the National Council for Scientific and Technological Development (CNPq), the Coordination for the Improvement of Higher Education Personnel (CAPES) (CAPES—Program CAPES/DFATD-88887.159028/2017-00, Program CAPES/COFECUB-88881.370894/2019-01) and the Foundation for Research Support of the State of Minas Gerais (FAPEMIG). Adriano de Oliveira Andrade is a fellow of CNPq, Brazil (304818/2018-6) and Luanne Cardoso Mendes, Isabela Alves Marques and Yann Morère are fellows of the Program CAPES/COFECUB (88887.612297/2021-00, 88887.628121/2021-01 and MA957/20 2019-2023, respectively). The funders had no role in study design, data collection and analysis, decision to publish, or preparation of the manuscript.

## Grant Disclosures

The following grant information was disclosed by the authors:

National Council for Scientific and Technological Development (CNPq): 304818/2018-6.

Coordination for the Improvement of Higher Education Personnel (CAPES): CAPES/DFATD-88887.159028/2017-00, Program CAPES/COFECUB-88881.370894/2019-01.

Foundation for Research Support of the State of Minas Gerais.

Program CAPES/COFECUB: 88887.612297/2021-00, 88887.628121/2021-01, MA957/20 2019-2023.

## Competing Interests

The authors declare there are no competing interests.

## Author Contributions

- Luanne Cardoso Mendes conceived and designed the experiments, performed the experiments, analyzed the data, performed the computation work, prepared figures and/or tables, authored or reviewed drafts of the article, contributed in the creation of the serious game narrative, designed the game phases, created the introductory videos explaining each phase, produced the game sound effects, and approved the final draft.
- Angela Abreu Rosa de Sá performed the experiments, analyzed the data, performed the computation work, prepared figures and/or tables, authored or reviewed drafts of the article, and approved the final draft.
- Isabela Alves Marques performed the experiments, analyzed the data, prepared figures and/or tables, authored or reviewed drafts of the article, and approved the final draft.
- Yann Morère performed the experiments, analyzed the data, performed the computation work, authored or reviewed drafts of the article, and approved the final draft.
- Adriano de Oliveira Andrade conceived and designed the experiments, performed the experiments, analyzed the data, performed the computation work, authored or reviewed drafts of the article, oriented and directed the team during the whole process of the game and interface device development, directly contributed in the evaluation of the system architecture, and approved the final draft.

## Ethics

The following information was supplied relating to ethical approvals (i.e., approving body and any reference numbers):

The study was approved by the Ethics Committee of the Federal University of Uberlândia under the CAAE number 39187620.6.0000.5152.

## Patent Disclosures

The following patent dependencies were disclosed by the authors:

The serious game described in this paper was registered at the Brazilian National Institute of Industrial Property (INPI –registration number: BR 51 2021 001975 0).

## Data Availability

Cardoso Mendes, Luanne, Alves Marques, Isabela, Abreu Rosa de Sá, Angela, Ramos Rosa, Rodrigo, Marques Alves, Camille, Rodrigues Lima, Kennedy, Afonso Lamounier Júnior, Edgard, Lázaro Martins Naves, Eduardo, Fraga Vieira, Marcus, Monteiro de Oliveira, Fábio Henrique, Pino, Pierre, Bourhis, Guy, Alves Pereira, Adriano, Tobias Machado, Carlos, Morère, Yann, & de Oliveira Andrade, Adriano. (2022). RehaBEElitation - Source code. Zenodo. https://doi.org/10.5281/zenodo.6985977

Cardoso Mendes, Luanne, Alves Marques, Isabela, Abreu Rosa de Sá, Angela, Ramos Rosa, Rodrigo, Marques Alves, Camille, Rodrigues Lima, Kennedy, Afonso Lamounier Júnior, Edgard, Lázaro Martins Naves, Eduardo, Fraga Vieira, Marcus, Monteiro de Oliveira, Fábio Henrique, Pino, Pierre, Bourhis, Guy, Alves Pereira, Adriano, Tobias Machado, Carlos, Morère, Yann, & de Oliveira Andrade, Adriano. (2022). RehaBEElitation serious game. Zenodo. https://doi.org/10.5281/zenodo.6988572

Cardoso Mendes, Luanne, Abreu Rosa de Sá, Angela, Alves Marques, Isabela, de Oliveira Andrade, Adriano, & Yann Morère. (2022, August 9). Introductory videos and videos showing the functioning of each phase of the RehaBEElitation serious game. Zenodo. https://doi.org/10.5281/zenodo.6977456

## Supplemental Information

Supplemental information for this article can be found online at http://dx.doi.org/10.7717/peerj-cs.1267#supplemental-information.

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
