# Peer review of "RehaBEElitation: the architecture and organization of a serious game to evaluate motor signs in Parkinson’s disease"

_PeerJ Computer Science, doi:10.7717/peerj-cs.1267_

## Round 0.1 · original submission · Major Revisions

Based on the reviewers comments, the authors are advised to make "Major Revisions" and resubmit.

·

Basic reporting

- Some errors in English style and grammar found.

Experimental design

- Well organized and good original primary research.

Validity of the findings

More experiments results for testing the accuracy of the game results may be appropriate.

Reviewer 2 ·

Basic reporting

The literature survey and related work sections are missing.
Please compare and contrast with prior artworks. It's difficult to identify the novelty of the architecture.

Experimental design

There is little information available on experimental design. The paper focused on the architecture and design implementation.

Validity of the findings

The research contributions are difficult to identify in the manuscript. The architecture and design implementation is not novel.

Additional comments

The article did not follow the research article format (literature and related work, main contributions, experimental validation, and comparison with prior artworks). Although the authors proposed architecture and implemented the SG design, the research contributions are difficult to identify.

·

Basic reporting

This paper needs major revision.

1- The introduction shows highlights the problem clearly.
2- The literature summary should be in a Table.
3- limitations should be in the conclusion section.
4- What is the novelty of this work?
4- Why need this approach? is there any draw back in the literature?

Experimental design

No comment

Validity of the findings

No comments

Additional comments

No comments

---

## Round 0.2 · Minor Revisions

The comments and suggestions of the reviewers has been well addressed. I only have some small comments to improve the paper.

(1) I suggest replacing the characterization of a SG given on line 90-91 by a better on, because as it is now given it more refers to gamification. My suggestion would be to use the definition given by Ritterfeld et al.: “any form of interactive computer-based game software for one or multiple players to be used on any platform and that has been developed with the intention to be more than entertainment” (Ritterfeld et al, 2009 , Page 6), or by Dörner et al. : “a digital game created with the intention to entertain and to achieve at least one additional goal (e.g., learning or health).”
U. Ritterfeld, M. Cody, P. Vorderer, Serious Games: Mechanisms and Effects, Routledge, 2009.
R. Dörner, S. Göbel, W. Effelsberg, J. Wiemeyer, Introduction, in: Serious Games, Springer, 2016, https://doi.org/10.1007/978-3-319-40612-1_1.

(2) ON and Off States: This is mentioned at several places in the paper (Abstract line 60,61, line 70; game accessibility evaluation line 663 -666) but it was not explained. Please introduce and explain them before referring to them.

(3) A paragraph at the end about future work would be nice

(4) The figures and tables seem to be missing in the new version of the paper

(5) Textual improvements:
Line 66-67: The heuristic evaluation  The heuristic usability evaluation
Line 139: to evaluate the system architecture  to evaluate the system’s usability
Line 140: we used evaluation  we did an evaluation
Line 143: may affect the patient’s evaluation  may affect the patent’s experience

---

## Round 0.3 · accepted · Accept

All comments have been addressed. I happy with the current version. It is ready for publication.